# Postprandial Lipid Metabolism in Normolipidemic Subjects and Patients with Mild to Moderate Hypertriglyceridemia: Effects of Test Meals Containing Saturated Fatty Acids, Mono-Unsaturated Fatty Acids, or Medium-Chain Fatty Acids

**DOI:** 10.3390/nu13051737

**Published:** 2021-05-20

**Authors:** Alexander Folwaczny, Elisa Waldmann, Julia Altenhofer, Kerstin Henze, Klaus G. Parhofer

**Affiliations:** Department of Internal Medicine 4, Ludwig-Maximilians University Munich, 81377 Munich, Germany; alex.folwaczny@googlemail.com (A.F.); elisa.waldmann@med.uni-muenchen.de (E.W.); julia.altenhofer@med.uni-muenchen.de (J.A.); kerstin.henze@med.uni-muenchen.de (K.H.)

**Keywords:** medium chain triglycerides, fatty acids, hyperlipidemia, MCT, hypertriglyceridemia

## Abstract

Fasting and postprandial hypertriglyceridemia are causal risk factors for atherosclerosis. The prevalence of hypertriglyceridemia is approximately 25–30% and most hypertriglyceridemic patients suffer from mild to moderate hypertriglyceridemia. Data regarding dietary interventions on postprandial triglyceride metabolism of mildly to moderately hypertriglyceridemic patients is, however, sparse. In a randomized controlled trial, eight mildly hypertriglyceridemic patients and five healthy, normolipidemic controls received three separate standardized fat-meals containing either saturated fatty acids (SFA), mono-unsaturated fatty acids (MUFA), or medium-chain fatty acids (MCFA) in a randomized order. Fasting and postprandial lipid parameters were determined over a 10 h period and the (incremental) area under the curve (AUC/iAUC) for plasma triglycerides and other parameters were determined. MCFA do not lead to a significant elevation of postprandial total plasma triglycerides and other triglyceride parameters, while both SFA (patients: *p* = 0.003, controls: *p* = 0.03 compared to MCFA) and MUFA (patients: *p* = 0.001; controls: *p* = 0.14 compared to MCFA) do lead to such an increase. Patients experienced a significantly more pronounced increase of plasma triglycerides than controls (SFA: patients iAUC = 1006 mg*h/dL, controls iAUC = 247 mg*h/dL, *p* = 0.02; MUFA: patients iAUC = 962 mg*h/dL, controls iAUC = 248 mg*h/dL, *p* = 0.05). Replacing SFA with MCFA may be a treatment option for mildly to moderately hypertriglyceridemic patients as it prevents postprandial hypertriglyceridemia.

## 1. Introduction

Fasting and postprandial hypertriglyceridemia (HTG) affects approximately 20% of the adult population in industrialized countries [1,2,3] and represents an established risk factor for atherosclerosis and acute pancreatitis [4,5,6,7]. Numerous studies indicate that subjects with elevated triglycerides have an increased risk for cardiovascular events compared to their respective normotriglyceridemic counterparts [6,8,9,10]. Data derived from Mendelian randomization studies indicate that this association is causal in nature [11,12]. Hypertriglyceridemia is usually classified on the basis of fasting triglyceride levels, although most people following a typical eating pattern of three or more meals a day are on average in a postprandial state for 20 out of 24 h [13,14]. Furthermore, studies suggest that postprandial hypertriglyceridemia in particular increases the risk of myocardial infarction [15,16]. In clinical practice, HTG is diagnosed if fasting triglycerides are above 150 mg/dL (1.7 mmol/L) and can be subdivided into mild to moderate HTG (150–1000 mg/dL; 1.7–11.4 mmol/L) and severe HTG (>1000 mg/dL; >11.4 mmol/L) [5].

For postprandial triglycerides, such cut-off values are not established, although postprandial triglycerides in normolipidemic subjects (unlike in HTG where massive increases can be observed) rarely exceed 400 mg/dL [5]. This lack of threshold values to diagnose and classify postprandial HTG also reflects the fact that the evaluation of postprandial lipid metabolism is not standardized and test meals differing in fat content, fat composition, and the content of other macronutrients (carbohydrates and/or protein) are used to evaluate postprandial lipid metabolism.

Considering that postprandial hypertriglyceridemia is a relevant residual risk factor for cardiovascular disease, it seems obvious to evaluate whether lipid modifying interventions also affect postprandial lipid metabolism. Such studies have been performed for statins [17,18], fibrates [19], ezetimibe [20], proprotein convertase subtilisin/kexin type 9 (PCSK9) inhibitors [21], and niacin [22,23], generally showing that these drugs affect postprandial lipid metabolism in a positive way. However, studies on the effect of dietary interventions on postprandial lipid metabolism are sparse.

Medium chain triglycerides (MCT) and medium chain fatty acids (MCFA; C-6–C-12) are metabolized differently from long-chain triglycerides (LCT) and long-chain fatty acids (LCFA; >C-14). While LCT are hydrolyzed in the intestine, “packed” into chylomicrons, and subsequently secreted into the lymph, from where they reach the circulation, MCFA directly enter portal blood and are transported to the liver [24]. MCT therefore cause significantly lower postprandial triglyceridemia than LCT and are used for prevention in HTG-induced acute pancreatitis and other gastrointestinal diseases like pancreatic insufficiency or short bowel syndrome [25]. While MCT are established in the treatment of severe HTG, little data is available for patients with mild to moderate HTG and healthy subjects.

We evaluated postprandial triglyceride metabolism after an oral fat-meal with either MCFA, saturated fatty acids (SFA) or monounsaturated fatty acids (MUFA) in normolipidemic subjects and patients with mild to moderate HTG.

## 2. Materials and Methods

The study was registered at ClinicalTrials.gov (NCT03846908) and performed between November 2018 and February 2020 at the University of Munich Medical Center.

Subjects: The study was planned to include five healthy normolipidemic subjects (controls) and ten mildly to moderately hypertriglyceridemic patients. Due to the COVID-19 pandemic, the study was terminated prematurely and only eight patients and five controls completed the study.

Patients were screened for mild to moderate HTG (inclusion criterion: casual fasting triglycerides 150–900 mg/dL (1.7–10.3 mmol/L) over a minimum period of three months and measured on at least two and up to four occasions; if triglycerides were outside the acceptable range at any time point, patients were not recruited). Key exclusion criteria were evidence of alcohol or drug abuse, diabetes mellitus, clinically relevant atherosclerotic disease, kidney (glomerular filtration rate < 60 mL/min) or liver disease of any etiology, uncontrolled thyroid disease or any other endocrine disease, acute or chronic inflammatory disease, any active malignancy, major surgical interventions within three months or planned, current or previous (3 months) treatment with antidiabetic or lipid lowering drugs, or a BMI over 35 kg/m^2^.

Healthy, normolipidemic controls were screened for normal fasting lipids (triglycerides 50–150 mg/dL; LDL-cholesterol < 190 mg/dL), with key exclusion criteria being any acute or chronic disease except medically controlled hypothyroidism (stable dose of L-thyroxine > 3 months) and any ongoing medication except contraception.

Study design: All participating subjects received three different isocaloric fat-meals on separate occasions in a randomized sequence (simple randomization, performed by KGP), spaced 7 to 28 days apart. Each fat-meal contained 80 g (absolute amount) of either MCT oil (Kanso^®^-nutrition. Dr Schär AG, Burgstall, Italy), SFA (Butaris^®^, Uelzena eG, Uelzen, Germany; commercially available butter fat), or MUFA (Rapso^®^, Rapso GmbH, Aschach, Austria; commercially available rapeseed oil), and was dissolved in a standardized “fat-shake” solution consisting of 100 mL water, 9 g de-oiled cacao-powder, one teaspoon of Stevia, and optionally 3.5 g of soluble coffee powder. Details of the test meals are shown in Table 1.

Subjects were blinded as to the fat-meal they received and no differences in taste were reported. Subjects were instructed to abstain from any alcohol consumption (for 2 days) and sports (for 1 day), and to consume an identical dinner in the evening prior to each test-meal in order to ensure comparable baseline values. During the 10 h testing period only water and unsweetened tea were consumed.

Methods: Using standard ethylenediaminetetraacetic acid (EDTA) and serum tubes (Sarstedt), blood samples were drawn in the fasting state (0 h) and after 1, 2, 4, 6, 8, and 10 h (EDTA; lipid parameters), and 1, 2, and 6 h (serum; insulin) after consumption of the fat-meal. Plasma/serum was obtained by centrifugation.

The analytical methods have been described in detail before [17]. In short, plasma samples were over-layered with a solution of d = 1.006 g/mL and ultra-centrifuged for 20 min at 20,000 rpm to isolate chylomicrons in the supranatant. Chylomicron triglycerides were then measured with a standard photometric kit. The infranatant was then again overlayered (d = 1.006 g/mL) and ultra-centrifuged for 18 h at 50,000 rpm in order to isolate chylomicron remnants (CR)/very low density lipoproteins (VLDL) in the supranatant and low density lipoproteins (LDL) as well as high density lipoproteins (HDL) in the infranatant. For ultracentrifugation, a Beckman Coulter ultracentrifuge was used.

Cholesterol, total triglycerides, apoB, and glucose were determined with standard photometric kits in plasma. Likewise, VLDL-cholesterol and CR/VLDL-triglycerides were measured in the supranatant, while HDL-and LDL-cholesterol were determined from the infranatant after precipitation with heparin and manganese chloride.

Postprandial metabolism was quantified by calculating the area under the curve (AUC) and the incremental AUC (iAUC) for plasma-, chylomicron-and CR/VLDL-triglycerides. Concentrations obtained over the 10-h period following the ingestion of the fat-meal were used for this calculation as described before [17,18]. The iAUC was determined as the area between the plasma concentration and a baseline or fasting concentration observed either at 0 h (if the 10 h value was greater or equal to the 0 h value) or 0 h and 10 h (if the 10 h value was smaller than the 0 h value). The iAUC represents the increase in area above fasting concentrations, resulting from the response to the fat load.

Statistical analysis: Sample size calculation indicated that 10 subjects are sufficient to identify significant differences between the MCT meal and the other fat-meals (effect size 0.9, power 0.8, α 0.5). Differences between parameters obtained during the test meals containing the different fats were evaluated by paired t-test analysis, while differences between the same parameters and the same fat-meal between mildly to moderately hypertriglyceridemic patients and normolipidemic subjects were evaluated by one way analysis of variance (ANOVA). Associations between variables were identified with the Pearson’s product moment correlation coefficient. Values are shown as mean ± standard deviation unless indicated otherwise. All statistical tests were performed using SPSS. Inc. software (SPSS. Inc., Chicago, IL, USA). The critical *p* value for significance was set at 0.05.

## 3. Results

The characteristics of the participating subjects and patients are shown in Table 2. Patients and controls differed significantly on parameters such as body mass index (BMI), fasting triglycerides and HDL-cholesterol.

Mean triglyceride levels over time following each meal are shown in Figure 1 (panel A normolipidemic subjects; panel B mildly to moderately hypertriglyceridemic patients). In both groups, on average no postprandial elevation of mean plasma triglyceride levels is observed if the test meal contains MCFA as fat, while comparable elevations of triglycerides are seen with test meals containing SFA or MUFA.

The mean incremental area under the curve (iAUC) for total triglycerides is shown in Figure 2 (panel A normolipidemic controls; panel B hypertriglyceridemic patients). For hypertriglyceridemic patients, the mean iAUC following the MCFA fat-meal is 152 mg*h/dL, and therefore significantly lower than for both SFA (1006 mg*h/dL; *p* = 0.03) and MUFA (962 mg*h/dL; *p* < 0.01). There is no significant difference between SFA and MUFA (*p* = 0.87). For normolipidemic controls, the mean iAUC following the MCFA fat-meal is 3 mg*h/dL and therefore lower than for both SFA (247 mg*h/dL; *p* = 0.025) and MUFA (248 mg*h/dL; *p* = 0.14). Again, there is no significant difference between SFA and MUFA (*p* = 0.91).

Hypertriglyceridemic patients peaked significantly higher than normolipidemic controls in terms of absolute plasma triglyceride levels and in terms of increase from 0 h baseline, but not in terms of percentage increase from 0 h baseline. When comparing hypertriglyceridemic patients to normolipidemic controls following both SFA and MUFA fat-meals, the iAUC for patients is significantly higher than for the normolipidemic controls (SFA: *p* = 0.019; MUFA: *p* = 0.05), while there is no significant difference for the mean iAUC for MCFA (*p* = 0.252).

Table 3 summarizes all parameters for hypertriglyceridemic patients and normolipidemic controls. No significant differences between an SFA and MUFA fat-meals were observed for either subject group. When comparing hypertriglyceridemic patients to normolipidemic controls following an MCFA fat-meal, we only found significant differences for the AUC of total triglycerides and CR/VLDL triglycerides and the absolute peak of triglycerides, while all iAUC parameters and the AUC for chylomicron triglycerides did not differ significantly.

Mean plasma chylomicron triglyceride levels over time (not shown) resemble those of total triglycerides on a smaller scale and the iAUC also roughly resembles the pattern described above for total triglycerides. Patients have a higher chylomicron triglyceride AUC than normolipidemic controls (significant for MUFA and approaching significance for SFA).

Total cholesterol, HDL-cholesterol, LDL-cholesterol, VLDL-cholesterol and apoB generally fluctuated slightly around fasting levels during the 10 h period.

While there is no significant difference (*p* = 0.15) for the HOMA-IR index between mildly to moderately hypertriglyceridemic patients and normolipidemic controls, patients on average had a HOMA-IR-index of 5.6 with a relatively high variability, while controls were in the normal range (Table 2). Following any fat-meal (SFA, MCFA, and MUFA), we observed a similar postprandial insulin peak after 1–2 h and a subsequent return to fasting levels. Glucose levels decreased slightly over the 10 h period in a similar fashion for both patients and controls.

Figure 3 shows fasting triglyceride levels for all subjects obtained prior to the three test-meals and during screening. The fasting triglyceride values for both hypertriglyceridemic patients and normolipidemic controls varied considerably.

Generally, subjects tolerated the fat-meals well, although some gastrointestinal symptoms such as soft or watery stool, stomach pain, or cramps and nausea were reported following the fat challenge. Symptoms were mostly mild and transient but were more frequent and pronounced for MCFA than for MUFA and SFA.

## 4. Discussion

In this study, we compared and quantified the effects of different LCT (SFA, MUFA) and MCT oil-meals on postprandial lipid metabolism for normolipidemic subjects and mildly to moderately hypertriglyceridemic patients under standardized conditions. Our results show that by substituting an experimental LCT fat-meal with an isocaloric MCT fat-meal, postprandial plasma triglycerides can be significantly reduced, in particular for patients with mild to moderate hypertriglyceridemia.

While both “healthy” (MUFA) as well as “unhealthy” (SFA) LCT-fat-meals lead to a sizeable and significant increase of postprandial plasma triglycerides, an MCT fat-meal does not cause any significant elevation of postprandial plasma triglycerides. This holds equally for both normolipidemic subjects and mildly to moderately hypertriglyceridemic patients. However, the effect of an LCT fat-meal is significantly more pronounced for hypertriglyceridemic patients than for normolipidemic controls: in mildly to moderately hypertriglyceridemic patients, AUC (“postprandial triglyceride load”) is about three times, while iAUC (“postprandial triglyceride elevation”) is approximately four times that of normolipidemic subjects. The “postprandial part” of hypertriglyceridemia, therefore, is much more pronounced than would intuitively be expected by looking at elevated fasting triglyceride levels alone. In contrast, after an MCT fat-meal there is no significant difference in the plasma triglyceride iAUC between normolipidemic subjects and mildly to moderately hypertriglyceridemic patients.

Our results are in line with the fact that LCT are metabolized differently from MCT, as MCFA bypass chylomicron formation and enter the portal bloodstream directly [24]. This is also reflected in the near zero iAUC for chylomicron triglycerides, in the significantly lower chylomicron AUC that we observed after an MCT oil-meal, and in a largely parallel chylomicron-and total-triglyceride curve (not shown).

Even patients with only mild to moderate fasting hypertriglyceridemia experience a considerable postprandial increase in plasma triglycerides over an extended period of time, after ingesting a standardized LCT fat-meal. This is not the case following an MCT fat-meal. The fact that patients had a higher AUC for total triglycerides and CR/VLDL triglycerides following the MCT fat-meal can be attributed to the higher fasting/baseline levels.

Fasting triglycerides also correlated significantly and positively with absolute postprandial increase in triglycerides but not with percentage increase for both LCT conditions (SFA: r = 0.77, *p* = 0.002; MUFA: 0.55, *p* = 0.05).

Figure 2 and Table 3 show that the iAUC for plasma triglycerides and most of the secondary parameters differ considerably between LCT (SFA and MUFA) on the one hand and MCFA on the other. In our normolipidemic subject group, significance can be demonstrated when comparing SFA to MCFA, but not when comparing MUFA to MCFA. This lack of significance may relate to the small sample size.

There is a consistent trend in the data of both subject groups for almost all examined triglyceride parameters to be smallest after an MCT fat-meal and largest after an SFA fat-meal. The values for triglyceride parameters after a MUFA fat-meal are mostly in between but tend to be more similar to the values of the SFA condition. However, as mentioned above, none of the differences between MUFA and SFA are significant.

The fact that HOMA-IR and fasting insulin where higher and more variable in the patient group indicates that some of the patients were insulin-resistant, which is to be expected in a group of patients suffering from mild to moderate hypertriglyceridemia. However, it should be noted that HOMA-IR is only a crude estimate of insulin sensitivity. The carbohydrate independent postprandial insulin response (similar for all three test meals) is probably mediated by the protein content of the test meals and gastro-intestinal hormones, the secretion of which were stimulated by the ingested volume.

By using standardized and isocaloric fat-meals, only differing in the type of fatty acid, we were able to demonstrate their differential postprandial triglyceride effects on triglyceride metabolism following a single pure-fat-meal. It is unclear whether regular consumption of MCT (replacement of LCT in daily diet) will only affect postprandial triglyceride levels or also decrease fasting triglyceride levels.

It is noteworthy that we observed a relatively high within subject variability of fasting triglyceride levels (Figure 3), given the fact that we made an effort to minimize/standardize potential known confounders such as sport, alcohol, and food consumption in the days prior to each test-meal.

A limitation of our study is the small subject number, which was further reduced by the COVID-19 pandemic. Due to the small subject number we are unable to provide data on subgroups (by gender, age, BMI, etc.) and more importantly can make no reliable statements on potential differences between MUFA and SFA. Furthermore, it is unclear whether the results would be identical if a mixed meal was used (instead of a pure fat challenge) and it is unclear (although likely) whether the results can be extrapolated to subjects with more severe hypertriglyceridemia. Finally, in the real world, meals are a mixture of different nutrients and interaction between fatty acids and other nutritional aspects (macro-nutrients or caloric balance) may limit the applicability of the study findings to clinical settings.

To conclude, mildly to moderately hypertriglyceridemic patients experience a considerable and, compared to normolipidemic controls, much more pronounced postprandial increase of plasma triglycerides after a single LCT fat-meal. This is not the case after an isocaloric MCT fat-meal. Patients with mild to moderate hypertriglyceridemia may therefore benefit from a substitution of LCT-fats with MCT-fats in their diet, but the mid-to long term effects of such a substitution on both fasting and postprandial lipid parameters should be explored in future trials.

## Figures and Tables

**Figure 1 nutrients-13-01737-f001:**
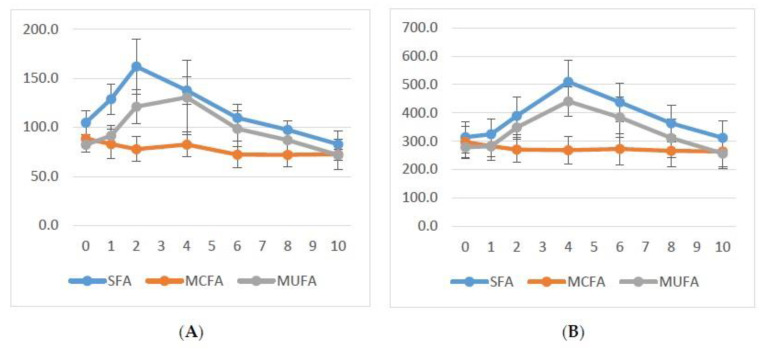
Mean total triglyceride levels (mg/dL) over time (h) (**A**) for five normolipidemic subjects and (**B**) for eight mildly to moderately hypertriglyceridemic patients. SFA: saturated fatty acids; MUFA: mono-unsaturated fatty acids; MCFA: medium-chain fatty acids. Shown are means and standard error of mean. Please note difference in scale on *y*-axis.

**Figure 2 nutrients-13-01737-f002:**
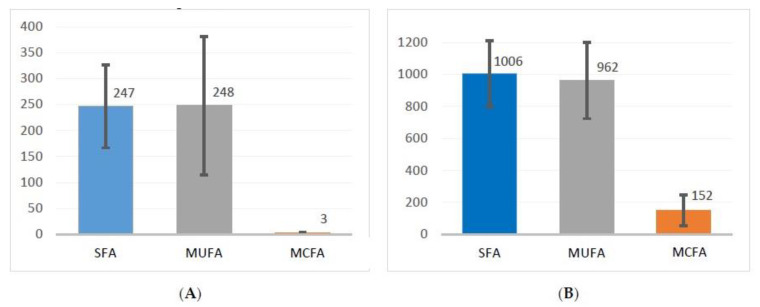
Mean incremental area under the curve (mg*h /dL) for total triglycerides (**A**) for five normolipidemic subjects and (**B**) for eight mildly to moderately hypertriglyceridemic patients. SFA: saturated fatty acids; MUFA: mono-unsaturated fatty acids; MCFA: medium-chain fatty acids. Shown are means and standard error of mean. Please note difference in scale on *y*-axis.

**Figure 3 nutrients-13-01737-f003:**
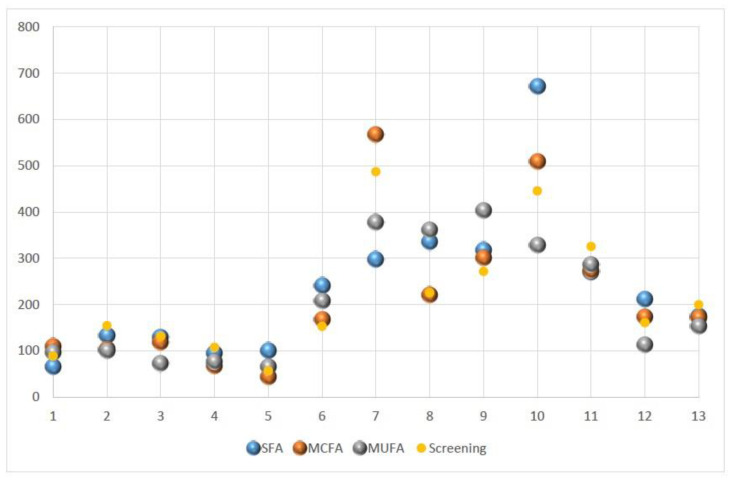
Variability in fasting triglyceride levels for five normolipidemic controls (subjects 1–5) and eight mildly to moderately hypertriglyceridemic patients (6–13) during screening and before each of the different test meals. SFA, saturated fatty acids; MCFA, medium chain fatty acids; MUFA, monounsaturated fatty acids.

**Table 1 nutrients-13-01737-t001:** Composition of test meals.

		Test Meals		
		SFA *	MUFA **	MCFA ***
Energy	kcal	742	735	735
Carbohydrates	g	0	0	0
Protein	g	1.4	1.2	1.2
Fat	g	81.2	81.2	81.2
Major FA				
SFA	%	68	8	99
MUFA	%	30	63	0
PUFA	%	2	29	0

* SFA-meal (27% palmitic acid, 22% oleic acid, 10% stearic acid, 1.8% linoleic acid,39% others). ** MUFA-meal (61.0% oleic acid, 20.0% linoleic acid, 9.0% α-linolenic acid, 10.0% others). *** MCFA-meal (58.4% caprylic acid, 38.3% capric-acid, 3.3% others). SFA, saturated fatty acids; MUFA, monounsaturated fatty acids; FA, fatty acids; MCFA, medium chain fatty acids.

**Table 2 nutrients-13-01737-t002:** Characteristics of the participating subjects and patients.

		Patients with Mild to Moderate Hypertriglyceridemia	Normolipidemic Subjects
N		8	5
Gender	m/f	6/2	2/3
Age	years	43 ± 12	35 ± 5
Body mass index	kg/m^2^	30 ± 4.0 **	23 ± 3.5
Fasting triglycerides	mg/dL	284 ± 118 **	108 ± 34
Total cholesterol	mg/dL	216 ± 37	186 ± 27
HDL-cholesterol	mg/dL	44 ± 12 **	68 ± 19
LDL-cholesterol	mg/dL	112 ± 35	96 ± 29
Lipoprotein(a)	mg/dL	14 ± 17	4.3 ± 1.9
Fasting glucose	mg/dL	96 ± 10	89 ± 3
HbA1c	%	5.2 ± 0.4 *	5.6 ± 0.26
C-reactive protein	mg/dL	0.43 ± 0.38	0.32 ± 0.27
Fasting insulin	µU/mL	23.9 ± 22.2	9.0 ± 2.6
HOMA-IR-index	-	5.6 ± 5.0	2.1 ± 0.59

Mean value ± standard deviation (SD) is shown; * trend, 0.1 ≥ *p* > 0.05; ** significant, *p* ≤ 0.05. HDL, high density lipoproteins; LDL, low density lipoproteins; HOMA-IR, homeostatic model assessment-insulin resistance.

**Table 3 nutrients-13-01737-t003:** Overview of parameters for mildly to moderately hypertriglyceridemic patients and normolipidemic subjects.

		Mildly to Mod. Hypertriglyceridemic Patients			Normolipidemic Subjects		
		SFA	MUFA	MCFA	SFA	MUFA	MCFA
iAUC triglycerides	mg*h/dL	1006 ± 583 ***	962 ± 673 ***	152 ± 271	259 ± 160 **^,++^	248 ± 298 *^,++^	3 ± 5
AUC triglycerides	mg*h/dL	4001 ± 1781 ***	3477 ± 1451 *	2716 ± 1412	1197 ± 276 *** ^+++^	1018 ± 299 ^+++^	770 ± 294 ^++^
iAUC chylomicron triglycerides	mg*h/dL	691 ± 396 ***	768 ± 488 ***	30 ± 75	231 ± 178 **^,++^	243 ± 286 ^+^	1 ± 1
AUC chylomicron triglycerides	mg*h/dL	1083 ± 680 ***	1025 ± 564 **	394 ± 503	427 ± 208**^,+^	377 ± 267 *^,++^	97 ± 71
iAUC CR/VLDL triglycerides	mg*h/dL	414 ± 323 *	351 ± 247 ***	126 ± 214	137 ± 129 *^,+^	115 ± 119 ^+^	29 ± 28
AUC CR/VLDL triglycerides	mg*h/dL	2989 ± 1303 **	2510 ± 1050	2323 ± 1045	716 ± 389 *^,+++^	521 ± 181 ^+++^	514 ± 307 ^+++^
Peak triglycerides (absolute)	mg/dL	520 ± 220 ***	474 ± 167 **	322 ± 175	171 ± 56 **^,+++^	150 ± 78 ^+++^	91 ± 31 ^++^
Increase triglycerides (peak-baseline)	mg/dL	205 ± 110 ***	196 ± 122 ***	24 ± 43	66 ± 51 **^,++^	68 ± 80 ^+^	2 ± 2
Relative increase triglycerides	%	72 ± 42 ***	81 ± 51 ***	7 ± 13	68 ± 54 *	88 ± 107	3 ± 3
Peak time triglycerides (Mean/Median/Mode)	h	4.8 ± 1.5/4/4 *	4.5 ± 1.8/4/4 *	2 ± 3.2/0/0	2 ± 1.2/2/1 ^+++^	4.8 ± 3/4/8	2.4 ± 2.2/4/4
Peak insulin (absolute)	µU/mL	35.0 ± 28.1	37.5 ± 39.8 **	44.8 ± 45.7	19.7 ± 10.1	12.3 ± 3.3 *	15.8 ± 5.3
Increase ins. (peak-baseline)	µU/mL	13.4 ± 10.4	12.1 ± 17.9	20.1 ± 20.7	10.3 ± 7.6	3.8 ± 3.0	6.7 ± 4.4

Unless otherwise stated, mean value ± SD is shown; asterisks (*) denote the level of significance for comparing either SFA or MUFA to MCFA (no significant differences were detected when comparing SFA to MUFA for any parameter); plus signs (+) denote the level of significance for comparing mildly to moderately hypertriglyceridemic patients to normolipidemic subjects. Auc, area under the curve; iAUC, incremental area under the curve; CR, chylomicron remnant; VLDL, very low density lipoprotein. * trend 0.1 ≥ *p* > 0.05; ** significant *p* ≤ 0.05; *** highly significant *p* ≤ 0.01; ^+^ trend 0.1 ≥ *p* > 0.05; ^++^ significant *p* ≤ 0.05; ^+++^ highly significant *p* ≤ 0.01.

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
