# Peer review of "Postprandial Lipid Metabolism in Normolipidemic Subjects and Patients with Mild to Moderate Hypertriglyceridemia: Effects of Test Meals Containing Saturated Fatty Acids, Mono-Unsaturated Fatty Acids, or Medium-Chain Fatty Acids"

_nutrients, 2021, doi:10.3390/nu13051737_

Round 1

Reviewer 1 Report

This study investigated the postprandial triglyceride(TG) metabolism after test fat meal (SFA, MUFA and MCFA) in subjects with mild to moderate hypertriglyceridemia and normal lipid profile. Study was done in carefully designed experimental setting to minimize confounding and the study findings were well-presented. Several minor points should be addressed before publication.

  1. It would be nice if you have data on the waist circumference and resent it in the Table 1.
  2. Triglyceride is well-known for its wide fluctuation with life habits such as diet, alcohol consumption and exercise. Though authors described that subjects were screened 'over a minimum period of three months and measured on at least two occasions', some more detail may be described. How many screening measurements were done? Were average of the measured TG used? Were measurement done casually or under controlled setting like during the application of the test meals?
  3. While the result of this experimental study clearly shows the differential effects of LCFA and MCFA, meals are mixture of all the nutrients in the real world. And those with hypertriglyceridemia (and obesity) are likely to be in positive caloric balance. There may be interaction between fatty acids and other nutrients, i. e., carbohydrate, and caloric balance. This may be another limitation of this study in applicability to clinical setting.

Author Response

We thank the reviewer for the thoughtful and constructive comments. We have addressed the topics in our revised manuscript.

Point by point answer:

  1. Unfortunately, we do not have data on waist circumference.
  2. Patients were recruited from out outpatient lipid clinic and were required to have elevated triglyceride levels over a minimum period of 3 months and measured on at least two occasions. Patients with one or more values outside the inclusion criteria were not recruited for the study (even if 2 other values were within the accepted range). Average values were not used. Lipid parameters were determined casually in the fasting state. This is clarified in the revised manuscript (methods section).
  3. Thank you, this is an important point and we have incorporated this in discussion section of the revised ms (“Finally, in the real world meals are a mixture of different nutrients and interaction between fatty acids and other nutritional aspects (macro-nutrients or caloric balance) may limit the applicability of the study findings to clinical settings.”).

Reviewer 2 Report

Authors have compared the effects of SFA, MUFA and MCFA on postprandial serum triglycerides in healthy and moderately hyper triglyceridemic human subjects. All study participant received each of the 3 test meals separated by 7-28 days. Compared with MCFA, SFA resulted in significant increase in postprandial TAG, while the results did not differ significantly between the effects of MCFA and MUFA. Postprandial hypertriglyceridemia was more pronounced in hyper triglyceridemic subjects than in those with normal fasting TAG. It is an interesting and timely study. Study design and interpretation seem appropriate. I would like the authors to address the following:

  1. Please provide the fatty acid composition of the test lipids used. This will help to clarify if the effects are caused by the form of the fatty acids or by individual fatty acid.
  2. Please state how the power calculations were made. N of 5 and 8 seem small.
  3. Clarify if the dose of the test fats used was same in all test subjects or it was adjusted for body weight.
  4. An increase in circulating free fatty acids (FFA) also increases the risk for NAFLD and CVD. It would have been desirable to determine the FFA concentrations as well in this study. It is good that MCFA decreased postprandial TAG, what effect did they have on FFA?
  5. Authors report no effects of the treatment meals on HOMA-IR. I recognize that it is a commonly used method but it is not very sensitive. Authors should make a note of it.

Author Response

We thank the reviewer for the thoughtful and constructive comments. We have addressed all topics in our revised manuscript.

  1. The revised manuscript includes a table in which the composition of the test meals is shown (methods section)
  2. Power calculations are clarified in the revised manuscript (methods section).
  3. Test meals were not adjusted for body weight. This is now clarified in the revised manuscript (methods section).
  4. This is an important point. Unfortunately, we have not measured free fatty acids in this trial.
  5. We have included a statement in the discussion section, that HOMA-IR is not a very sensitive method when evaluating the effect of food components on insulin sensitivity.